# Validation of Inactivation Methods for Arenaviruses

**DOI:** 10.3390/v13060968

**Published:** 2021-05-24

**Authors:** Silke Olschewski, Anke Thielebein, Chris Hoffmann, Olivia Blake, Jonas Müller, Sabrina Bockholt, Elisa Pallasch, Julia Hinzmann, Stephanie Wurr, Neele Neddersen, Toni Rieger, Stephan Günther, Lisa Oestereich

**Affiliations:** 1Department of Virology, Bernhard-Nocht Institute for Tropical Medicine, 20359 Hamburg, Germany; olschweski@bnitm.de (S.O.); thielebein@bnitm.de (A.T.); hoffmann@bnitm.de (C.H.); olivia.blake@bnitm.de (O.B.); jonas.mueller@bnitm.de (J.M.); bockholt@bnitm.de (S.B.); pallasch@bnitm.de (E.P.); hinzmann@bnitm.de (J.H.); wurr@bnitm.de (S.W.); neddersen@bnitm.de (N.N.); rieger@bnitm.de (T.R.); guenther@bnitm.de (S.G.); 2German Center for Infectious Research (DZIF), Partner Site Hamburg-Lübeck-Borstel-Riems, 20359 Hamburg, Germany

**Keywords:** high-risk pathogens, arenaviruses, inactivation

## Abstract

Several of the human-pathogenic arenaviruses cause hemorrhagic fever and have to be handled under biosafety level 4 conditions, including Lassa virus. Rapid and safe inactivation of specimens containing these viruses is fundamental to enable downstream processing for diagnostics or research under lower biosafety conditions. We established a protocol to test the efficacy of inactivation methods using the low-pathogenic Morogoro arenavirus as surrogate for the related highly pathogenic viruses. As the validation of chemical inactivation methods in cell culture systems is difficult due to cell toxicity of commonly used chemicals, we employed filter devices to remove the chemical and concentrate the virus after inactivation and before inoculation into cell culture. Viral replication in the cells was monitored over 4 weeks by using indirect immunofluorescence and immunofocus assay. The performance of the protocol was verified using published inactivation methods including chemicals and heat. Ten additional methods to inactivate virus in infected cells or cell culture supernatant were validated and shown to reduce virus titers to undetectable levels. In summary, we provide a robust protocol for the validation of chemical and physical inactivation of arenaviruses in cell culture, which can be readily adapted to different inactivation methods and specimen matrices.

## 1. Introduction

Pathogenic RNA viruses have repeatedly led to severe outbreaks including the influenza virus pandemic that started in 1918, the human immunodeficiency virus 1 (HIV) global epidemic, the Ebola virus (EBOV) outbreak in 2014, the Zika virus pandemic in 2016 and the current severe acute respiratory syndrome coronavirus type 2 (SARS-CoV-2) pandemic. Zoonotic viruses are responsible for the great majority of the recently emerging infectious diseases. The ongoing SARS-CoV-2 pandemic highlights the importance of fast and sensitive diagnostic methods to enable contact tracing and isolation, to prevent the spread of the disease. Diagnostics and research with high-risk pathogens, however, can be challenging, as pathogen-specific biosafety measures need to be adhered to. A rapid and safe inactivation is very often a basic requirement for many diagnostic and research methods to allow further processing under lower biosafety conditions. The arenaviruses are enveloped segmented single-strand RNA viruses that are distributed worldwide and cause annual zoonotic outbreaks. Several arenaviruses like Machupo virus (MACV), Junín virus (JUNV), Guanarito virus, Sabiá virus, and Lassa virus (LASV) can cause hemorrhagic fevers in humans and thus are a serious public health concern. LASV especially has epidemic potential due to the high number of annual cases and the lack of antivirals or a vaccine and is therefore listed in the WHO R&D Blueprint, which contains a list of potential epidemic threats needing urgent R&D action [1].

Work with human pathogenic arenaviruses including LASV requires the highest level of biosafety precautions and only a few laboratories worldwide are suitably equipped to conduct such research. Handling samples containing these viruses outside the high biosafety containment requires sample inactivation, which is mostly validated in-house. So far, successful inactivation of arenaviruses with TRIzol, Formalin [2,3], gamma irradiation [4,5,6,7], a photoactive compound in combination with UV irradiation [8,9,10], pH and heat [6,11] have been published. Many other common methods for viral inactivation including acetone or detergents such as Triton X-100 have not been tested. The inactivation efficacy of the frequently used guanidine thiocyanate-containing lysis buffers for RNA purification has only been evaluated for other RNA viruses like EBOV [12,13].

The validation of inactivation methods is difficult as they often include cytotoxic chemicals, which makes it challenging to demonstrate the loss of infectivity of a potential inactivated sample *in vitro* without causing excessive cytotoxicity. Therefore, we have developed a validation protocol to assess the inactivation efficiency of different chemicals. We used Morogoro virus (MORV), a risk group 2 pathogen closely related to LASV, as a surrogate for the *Arenaviridae* family. The developed protocol was validated with well-established viral inactivation methods including heat and Formalin. Applying our validation protocol, we evaluated the efficiency of different commonly used inactivation methods. In total, we tested 12 methods, 7 for infected cells and 5 for virus-containing fluids, which allow subsequent RNA isolation, serology analysis, protein analysis using Western blot, immune fluorescence microscopy, histology and flow cytometry.

## 2. Materials and Methods

### 2.1. Cells and Virus Stocks

The amplification of the virus stock, the production of infected cells, and the monitoring of infectivity of potentially inactivated samples was performed in VERO 76 cells (ATCC^®^ CRL-1587™, American Type Culture Collection, Manassas, USA). The cells were maintained in medium containing Dulbecco’s minimal essential medium (DMEM) supplemented with 3% fetal calf serum (FCS), 100 U/mL penicillin, 100 µg/mL streptomycin, 1 mM glutamine, 0.5 mM pyruvate, and 1× non-essential amino acids (all from Pan Biotech, Aidenbach, Germany) at 37 °C and 5% CO_2_.

MORV strain 3017/2004 had been isolated and sequenced in our laboratory [14] and the used stock was passaged ≤3 times. For the amplification of the stock, cells were infected with a multiplicity of infection (MOI) of 0.01, the virus-containing supernatant was harvested three days post infection and filtered through a 0.1 µm sterile filter unit (Millipore, Burlington, MA, USA). It was further concentrated to 5 × 10^6^ foci forming units (FFU) per mL via ultrafiltration (see below). The viral titer was determined by immunofocus assay (IFA) as described elsewhere [15,16,17], and the stock was stored at −80 °C until use. This concentrated stock was employed for the validation of the efficacy of the different inactivation methods. For the validation of the efficacy of cellular inactivation methods, cells were infected with a MOI of 0.01 and harvested three days post infection. The supernatant was removed and the cells were trypsinized until detachment, which was monitored visually. The infection rate of the cells was determined by indirect immunofluorescence staining [18] and cells were only used if an infection rate above 50% was reached. In IFA as well as in indirect immunofluorescence staining, MORV was detected with the Old-World arenavirus NP-specific monoclonal antibody 2LD9 [19].

### 2.2. Viral Concentration

The recovery rate of infectious arenaviruses with the Amicon^®^Ultra-15 100K Centrifugal Filter Device (Millipore, Burlington, MA, USA) as concentrator was evaluated for MORV. Fifty mL of viral stocks ranging from high to low amounts of infectious particles (10^6^ to 100 FFU) were centrifuged through the device at room temperature (RT) until the volume was reduced to 2 mL in triplicates according to the manufacturer’s specifications. The amount of infectious viral particles in the concentrate was analyzed by IFA to determine the recovery rate. Additionally, the amount of viral particles in the input and the flow-through were determined. Viral RNA in input, concentrate, and flow-through was quantified with MORV-specific real-time RT-PCR (see Appendix A).

The filter device was also used for the buffer exchange of the inactivated MORV samples to remove cytotoxic agents. To prevent damage of the filters due to the reagents which would result in higher permeability of the filter, all samples were diluted to reagent concentrations below the manufacturer’s specified limits.

### 2.3. Inactivation of Infectious Material

All validation experiments were performed in independent triplicates. The negative controls without virus were treated similarly as the samples, while for the positive controls, all reagents were replaced by equivalent volumes of 1× PBS, and incubations were performed at RT. For the validation of methods for the inactivation of infected cells, triplicates of 2 × 10^7^ MORV-infected cells (infection rate > 50%) were used. As a negative control, 1 × 10^6^ not-infected cells were inactivated and, as a positive control, 2 × 10^6^ MORV-infected cells (infection rate > 50%) were used. The validation of methods for the inactivation of infected supernatant was performed with 200 µL virus stock, equivalent to 1 × 10^6^ FFU of MORV. As a negative control, the cell culture supernatant of mock infected cells was used. A summary of the investigated methods is given in Table 1, and the inactivation procedures are described in the following paragraphs.

Vero cells were infected with the inactivated samples or controls and cultured in total for 28 days. The cells were visually inspected on a daily basis and split 1:10 once or twice per week depending on their growth rate. The medium of the flasks was kept and added to the new flask in a ratio of 1:2 to maintain the potentially infectious virus in the supernatant. To monitor the infection rate of the cells, samples for immunofluorescence staining and IFA were collected weekly. A detailed scheme of the complete protocol is given in Figure 1.

#### 2.3.1. Tested Inactivation Methods for Infected Cells without Cell Lysis

The infected cells were pelleted by centrifugation for 5 min at 500× *g* and subsequently resuspended in 2 mL acetone (100%), acetone/methanol (1:1), methanol (100%), 4% formaldehyde in 1× PBS (all Carl Roth, Karlsruhe, Germany), or BD Cytofix/Cytoperm™ buffer (BD BioScience; San Jose, USA). The cells were incubated for 20 min (acetone, acetone/methanol, methanol) or 30 min (4% formaldehyde, BD Cytofix/Cytoperm™) at RT. Afterwards, the cells were washed three times with 50 mL 1× PBS and pelleted at 300× *g* for 5 min. The cells were resuspended in 15 mL medium and used to infect pre-seeded cells.

#### 2.3.2. Tested Inactivation Methods for Infected Cells with Cell Lysis

To test the inactivation with Buffer AL from the DNeasy Blood and Tissue kit (Qiagen, Hilden, Germany), the infected cells were pelleted (centrifugation for 5 min at 500× *g*) and resuspended in 0.8 mL 1× PBS. 80 µL proteinase K and 0.8 mL buffer AL were added followed by incubation at 56 °C for 10 min (procedure according to manual). For the testing of SDS sample buffer (containing 6% natriumlaurylsulfat (SDS), 150 mM tris(hydroxymethyl)aminomethane pH 6.8, 30% glycerol, 100 mM dithiothreitol, bromphenolblue), the pelleted cells were resuspended in 1 mL 1× PBS, mixed with 0.5 mL SDS sample buffer, and heated for 10 min at 95 °C on a heat block.

The inactivated samples from the two methods were afterwards diluted to 100 mL in 1× PBS to reduce the concentration of chemicals in the samples below the manufacturer’s specified limits of the Amicon^®^Ultra-15 100K Centrifugal Filter Device. They were concentrated to 1 mL by centrifugation with up to 2500× *g* for 2 to 10 min. The concentrate was again diluted in 10 mL 1× PBS and concentrated to 1 mL. This concentrate was diluted in medium to a total volume of 15 mL and used to infect pre-seeded cells.

#### 2.3.3. Tested Inactivation Methods for Infectious MORV in Cell Culture Supernatant

##### AVL Buffer, Triton X-100 and SDS Buffer

Note that 800 µL of buffer AVL from the QIAamp Viral RNA Mini Kit (Qiagen, Hilden, Germany) were mixed with 200 µL of MORV stock and incubated for 10 min at RT. Afterwards, 800 µL of >99% ethanol (Carl Roth, Karlsruhe, Germany) were added to the AVL-virus mixture.

200 µL of a solution of 2% Triton X-100 in 1× PBS (Carl Roth, Karlsruhe, Germany) were mixed with 200 µL of MORV stock and incubated for 30 min at RT. 

100 µL of SDS sample buffer (see above) were mixed with 200 µL of MORV stock and incubated for 10 min at 95 °C on a heat block

The inactivated samples were diluted to 50 mL in 1× PBS to reduce the concentration of chemicals in the samples below the manufacturer’s specified limits of the Amicon^®^Ultra-15 100K Centrifugal Filter Device. They were concentrated to 1 mL by centrifugation with up to 2500× *g* for 2 to 10 min. The concentrate was again diluted in 10 mL 1× PBS and concentrated to 1 mL. This concentrate was diluted in medium to a total volume of 15 mL and used to infect pre-seeded cells.

##### Heat Inactivation

For heat inactivation, 200 µL of the MORV stock was incubated for 60 min at 60 °C on a heat block. It was afterwards diluted in 15 mL medium and used to infect pre-seeded cells.

##### Serobuvard Filter Paper

For the inactivation of MORV on Serobuvard filter paper (Serobuvard, LDA 22, Zoopole, France), 200 µL of MORV stock were spotted on a small piece of filter paper and dried for 24 h and RT. The dried spot was placed into a conical centrifuge tube, and 15 mL of medium were added. The tube was agitated for 1 h at 37 °C and 1000 rpm. The medium was used to infect pre-seeded cells.

## 3. Results

### 3.1. Verification of Ultrafiltration to Concentrate Arenaviruses

The use of the Amicon^®^Ultra-15 Centrifugal Filter Device as concentrator has been described for a large variety of viruses and just recently also for arenaviruses such as JUNV, MACV, LASV and Tacaribe virus (TCRV) [21,22]. However, the recovery rate of these devices when used for concentrating arenaviruses has not yet been determined, and published recovery rates of infectious particles vary between 90% and 100% [23] for retroviruses to only 15% for SARS-CoV-2 [24]. Therefore, we first determined the recovery rate of infectious virus from cell culture supernatant for the arenavirus MORV.

To assess the impact of the virus concentration in the starting material on the recovery rate, we performed ultrafiltration with different input amounts of MORV. The numbers of infectious particles ranged from 10^6^ to 100 FFU, and the experiments were done in triplicates. The mean recovery rate of infectious MORV was found to be 56–106%, and the recovery rate of the viral RNA ranged from 49 to 117% (Figure 2). Lower recovery rates were generally observed for the lower input amounts. However, we still achieved more than 50% mean recovery at the lowest tested input (100 FFU). Therefore, ultrafiltration with this device is well suited to concentrate arenaviruses and was used in this study to remove cytotoxic substances after inactivation.

### 3.2. Evaluation of the Efficacy of Tested Inactivation Methods

We first tested our developed inactivation validation protocol with the published methods of inactivation using 4% formaldehyde and heat as a proof-of-concept experiment. We could replicate the previously published results of complete inactivation of infectious particles with both methods. Next, we validated the other 10 methods and successfully demonstrated complete inactivation for each method. We could not detect infectious MORV by immunofluorescence staining or IFA within 4 weeks post-inoculation in any of the negative controls or the inactivated triplicates. Infectious MORV was, however, detected at all investigated time points in the positive controls with both control assays. Therefore, we could show that all tested methods lead to a complete virus inactivation. A summary of the results is given in Table 2.

## 4. Discussion

Diagnostics and research of high-risk pathogens in lower biosafety environments depends on the availability of safe and efficacious inactivation methods. While numerous comprehensive inactivation studies have been published for some high-risk viruses such as EBOV [12,13,25,26,27], this has so far been lacking for the *Arenaviridae* family. As a result, working with arenaviruses often requires a time-consuming in-house validation process of inactivation methods. One aim was to develop an *in vitro* protocol that allows researchers to assess the inactivation efficacy of different commonly available reagents and treatments for non-toxic as well as for cytotoxic reagents. Before samples inactivated with cytotoxic reagents come into contact with cells, the respective substances must be reduced or removed either by dilution and subsequent concentration or by buffer exchange. Buffer exchange, for example by dialysis, leads to a significantly increased incubation time in the inactivating agents, which can, by itself, reduce the number of infectious viruses and may falsify the results. Hence, we used Amicon^®^Ultra-15 100K Centrifugal Filter Devices to concentrate arenaviruses via ultrafiltration and to remove cytotoxic substances. We observed a high mean recovery rate of infectious MORV particles of 56–106%, which is close to the recovery rate for retroviruses [23]. The recovery rate of infectious MORV particles as well as the recovery rate of viral RNA showed a dependency on the amount of virus. However, even inputs as low as 100 FFU showed a mean recovery rate above 50%. Therefore, ultrafiltration represents a suitable tool to generate high-titer arenaviral stocks or to study samples with low virus concentrations, as in the case of waste-water studies. In contrast to other studies [13,28,29], we had a low volume of the inactivated samples after the concentration preventing dilution effects, which allowed us to inoculate the complete inactivated sample on one T-75 flask of cells. We also chose a very long passaging time of four weeks. Both measures ensured that even the smallest amount of possibly non-inactivated virus could amplify.

With this study, we provide a protocol to assess the efficacy of a broad range of inactivation methods for the most widely used specimens in laboratory research (cell culture supernatant and infected cells). Using our protocol, we validated a broad range of inactivation methods for viruses of the *Arenaviridae* family. We could confirm the previously shown complete inactivation with heat and with 4% formaldehyde. Furthermore, we validated other commonly used inactivation methods including AVL, 1% Triton X-100, SDS-buffer, Serobuvard filter paper, acetone, methanol, AL buffer and BD Cytofix/Cytoperm™. All 10 additional tested inactivation methods completely inactivated the infectious cell culture supernatant and infected cells and thus allow the downstream processing in numerous applications such as flow cytometry, serology, or RNA isolation in laboratories with lower biosafety level. Our study focused on the inactivation of infected cells and cell culture supernatant and the inactivation efficacy may vary for other specimens such as serum, plasma, stool, saliva, or tissue. For EBOV, it has already been shown that detergent-based inactivation was less efficient in serum compared to cell-culture medium [20]. Therefore, inactivation experiments may have to be repeated using downstream application-specific matrices. We expect that our protocol can be easily adapted to other matrices and used for the validation of further inactivation methods.

## Figures and Tables

**Figure 1 viruses-13-00968-f001:**
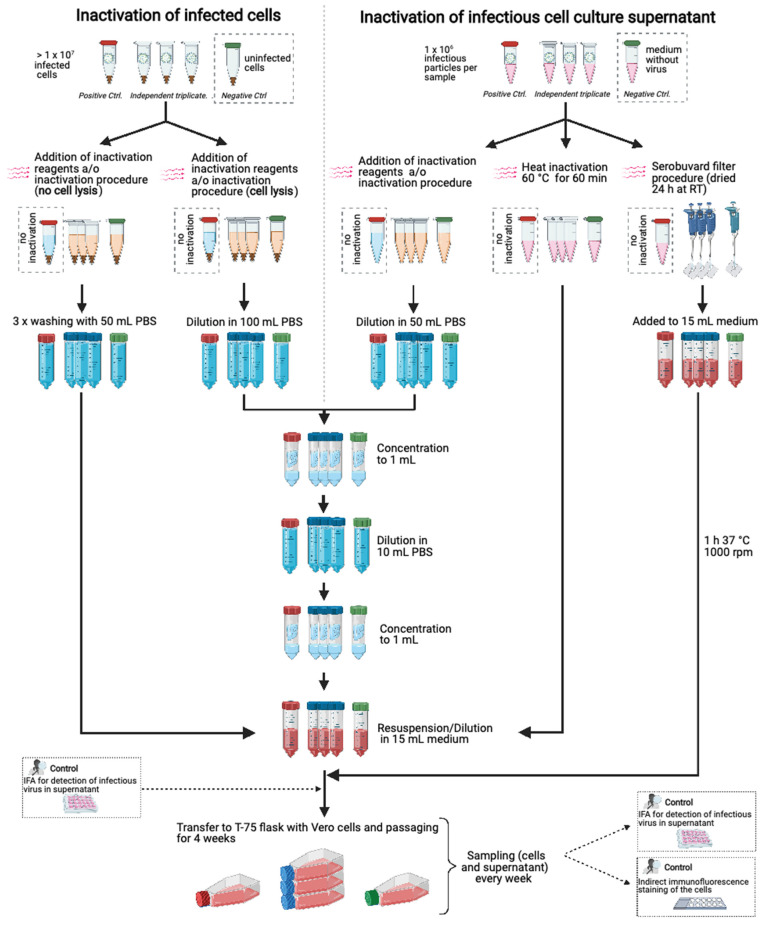
Scheme of the protocol for testing inactivation efficacy in vitro. For the validation of methods to inactivate MORV infected cells, triplicates with 2 × 10^7^ cells, a positive control with 2 × 10^6^ cells (infection rate for both >50%) and 10^6^ uninfected cells as a negative control were used. After the inactivation procedure for methods without cell lysis, the cells were washed three times with 50 mL PBS and resuspended in medium. For methods where cells lysed during the inactivation, samples were diluted in 100 mL PBS, concentrated to 1 mL, diluted to 10 mL and concentrated again to 1 mL. For the validation of methods to inactivate MORV-containing cell culture supernatant, 1 × 10^6^ infectious particles were used for the testing approach and the positive control while the negative control contained virus-free medium. After the inactivation procedures, the samples were diluted to 50 mL in PBS, concentrated to 1 mL, diluted to 10 mL with PBS and concentrated to 1 mL. The concentrates of the different methods were diluted in medium (15 mL total volume). Heat-inactivated supernatant was diluted in medium directly after incubation, while for the inactivation on Serobuvard paper virus was spotted, dried for 24 h and added to medium for 1 h at 37 °C and 1000 rpm. The resuspended cells and diluted samples were used to inoculate pre-seeded cells which were cultured for 4 weeks after infection. The infection rate of cells and supernatant was monitored by indirect immunofluorescence staining and IFA. This figure has been created with BioRender.com.

**Figure 2 viruses-13-00968-f002:**
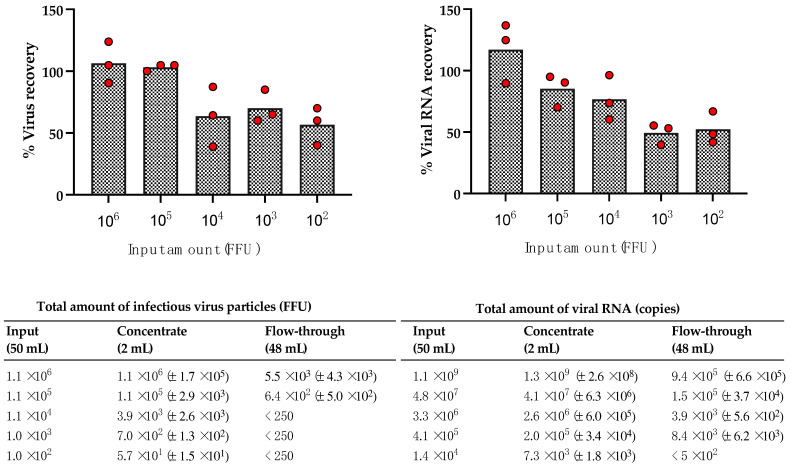
Viral recovery rates after concentration of MORV with Amicon^®^Ultra-15 Centrifugal Filter Devices depending on the input concentrations. Different amounts of MORV virus stock ranging from 10^6^ FFU to 100 FFU were diluted in 50 mL and concentrated to 2 mL using Amicon^®^Ultra-15 Centrifugal Filter Devices in triplicates for each input concentration. The amounts of infectious viral particles in the input (50 mL), concentrate (2 mL), and flow-through (48 mL) were determined with IFA and used to calculate the recovery rate. Recovery rates of viral RNA were determined with MORV-specific real-time RT-PCR. The recovery rates in the concentrate for each experiment are depicted as red circles and the bars represent the mean. The amounts of infectious viral particles as well as viral RNA amount for the input, the concentrate and the flow-through are shown in the table. Values are given as mean with standard deviation.

**Table 1 viruses-13-00968-t001:** Overview of tested inactivation methods.

Inactivation Method	Purpose	Specimen	Inactivation Conditions
100% Acetone	Immune fluorescence microscopy	Infected cells, without lysis	RT, 20 min
100% Methanol	Immune fluorescence microscopy	Infected cells, without lysis	RT, 20 min
Acetone/methanol (1:1)	Immune fluorescence microscopy	Infected cells, without lysis	RT, 20 min
4% Formaldehyde	Immune fluorescence microscopy, histology, flow cytometry	Infected cells, without lysis	RT, 30 min
BD Cytofix/Cytoperm™	Flow cytometry	Infected cells, without lysis	RT, 30 min
AL buffer + proteinase k + ethanol	DNA isolation	Infected cells, with lysis	50 °C, 10 min
SDS buffer + heat	Western Blot	Infected cells, with lysis	95 °C, 10 min
AVL buffer + ethanol	RNA isolation	Cell culture supernatant	RT, 10 min
1% Triton x-100	Serology ^a^	Cell culture supernatant	RT, 30 min
SDS buffer + heat	Western Blot	Cell culture supernatant	95 °C, 10 min
Heat	Serology	Cell culture supernatant	60 °C, 60 min
Serobuvard filter paper	Serology RNA isolation	Cell culture supernatant	RT, 24 h

Abbreviations: RT = room temperature; min = minute; h = hour; ^a^ Note that the present study has validated detergent-based inactivation of virus only for serum samples pre-diluted at least 1:20 in buffer (serum concentration <5%) before addition of detergent. This concentration corresponds to the FCS concentration in our test matrix (cell culture supernatant). As shown for Ebola virus, high serum concentration may affect inactivation efficacy of detergents [20].

**Table 2 viruses-13-00968-t002:** Overview of the efficacy of the tested inactivation methods.

Inactivation Method	Sample Type	Quantity of Infectious Material	Result (Successful/Total)
100% Acetone	Infected cells	>10^7^ cells infected cells	Complete inactivation (3/3)
100% Methanol	Infected cells	>10^7^ cells infected cells	Complete inactivation (3/3)
Acetone/methanol (1:1)	Infected cells	>10^7^ cells infected cells	Complete inactivation (3/3)
4% Formaldehyde	Infected cells	>10^7^ cells infected cells	Complete inactivation (3/3)
BD Cytofix/Cytoperm™	Infected cells	>10^7^ cells infected cells	Complete inactivation (3/3)
AL buffer + proteinase k + ethanol	Infected cells	>10^7^ cells infected cells	Complete inactivation (3/3)
SDS buffer + heat	Infected cells	>10^7^ cells infected cells	Complete inactivation (3/3)
AVL buffer + ethanol	Cell culture supernatant	10^6^ FFU	Complete inactivation (3/3)
1% Triton x-100	Cell culture supernatant	10^6^ FFU	Complete inactivation (3/3)
SDS buffer + heat	Cell culture supernatant	10^6^ FFU	Complete inactivation (3/3)
Heat	Cell culture supernatant	10^6^ FFU	Complete inactivation (3/3)
Serobuvard filter paper	Cell culture supernatant	10^6^ FFU	Complete inactivation (3/3)

## Data Availability

Not applicable.

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
