# Peer review of "Validation of Inactivation Methods for Arenaviruses"

_viruses, 2021, doi:10.3390/v13060968_

Round 1

Reviewer 1 Report

Overall, this is a nice paper that provides validation for many strategies to inactivate arenaviruses.  There are a few issues that need to be addressed, however.

Major:

  1. Inactivation methods for serum/serology can not be validated in cell culture medium. Inactivation in samples for assays using serum and whole blood need to be performed in serum and whole blood.  This was shown with EBOV in the following publication.  van Kampen, JJA et al. Ebola Virus Inactivation by Detergents Is Annulled in Serum. JID 216(7) Oct. 1, 2017. 859-866 DOI: 10.1093/infdis/jix401.  Misinterpretation of the results in this manuscript i.e that if it works in cell culture medium it will work for serum or whole blood especially if the purpose is listed as "Serology" in the manuscript, could have disastrous results 
  2. I couldn’t follow the conclusions in the results section 3.1. The paper states that the authors started with 50 mL of 1.4 X 106 FFU/mL MORV and concentrated it to 2 mL of 5.5 X 107 FFU/mL and achieved 85% recovery.  To me this means: 1.4 X 106 X 50 = 7 X 107 was the initial amount of virus and 5.5 X 107 X 2 = 1.1 X 108 was the amount of virus in the concentrate.  That is an increase of 157%.  The results with LASV were similar.  Either I didn’t understand what was done or there is something wrong with the math.

Minor:

Line 38:  This is not the family name so it shouldn’t be capitalized.

Line 39:  Please change annually to annual.

Line 30 & 40:  Be consistent with the shorthand virus names Guanirito virus and Sabiá virus.

Line 75:  Please indicate how the virus stock was validated, for instance sequencing, and or a reference to how the virus was identified.

Line 134:  Please change volumn to volume.

Line 225:  Please change Formaldehyde to formaldehyde

Author Response

Please find enclosed our reply to the comment from Reviewer 1. All changes in the manuscript are highlighted in yellow (named “viruses-1187493_changes highlighted”) and one additional Appendix A has been added.

Overall, this is a nice paper that provides validation for many strategies to inactivate arenaviruses. There are a few issues that need to be addressed, however.

Major:

Inactivation methods for serum/serology can not be validated in cell culture medium. Inactivation in samples for assays using serum and whole blood need to be performed in serum and whole
blood. This was shown with EBOV in the following publication. van Kampen, JJA et al. Ebola Virus Inactivation by Detergents Is Annulled in Serum. JID 216(7) Oct. 1, 2017. 859-866 DOI: 10.1093/infdis/jix401. Misinterpretation of the results in this manuscript i.e that if it works in cell culture medium it will work for serum or whole blood especially if the purpose is listed as "Serology" in the manuscript, could have disastrous results.

Thank you very much for this comment. We changed the discussion accordingly to reflect the impact different matrices can have on inactivation efficiency. We changed the last paragraph in the discussion: “All 10 additional tested inactivation methods completely inactivated the infectious samples cell culture supernatant and infected cells and thus allow the downstream processing in numerous applications such as flow cytometry, serology, or RNA isolation in laboratories with lower biosafety level. Our study focused on the inactivation of infected cells and cell culture supernatant and the inactivation efficacy may vary for other specimens such as serum, plasma, stool, saliva, or tissue. For EBOV it has already been shown that detergent-based inactivation was less efficient in serum compared to cell-culture medium [20]. Therefore, inactivation experiments may have to be repeated using downstream application-specific matrices. We expect that our protocol can be easily adapted in the future to other matrices and used for the validation of further inactivation methods.”

Member of the Leibniz Association
National Reference Centre for Tropical Pathogens
WHO Collaborating Centre for Arbovirus and Haemorrhagic Fever Reference and Research www.bnitm.de

Tel Fax E-Mail

To further emphasis that the inactivation of serum for serology using detergent-based buffers might be impacted, we added a footnote to table 1: “Note that the present study has validated detergent-based inactivation of virus only for serum samples pre-diluted at least 1:20 in buffer (serum concentration <5%) before addition of detergent. This concentration corresponds to the FCS concentration in our test matrix (cell culture supernatant). As shown for Ebola virus, high serum concentration may affect inactivation efficacy of detergents [20].”

I couldn’t follow the conclusions in the results section 3.1. The paper states that the authors started with 50 mL of 1.4 X 106 FFU/mL MORV and concentrated it to 2 mL of 5.5 X 107 FFU/mL and achieved 85% recovery. To me this means: 1.4 X 106 X 50 = 7 X 107 was the initial amount of virus and 5.5 X 107 X 2 = 1.1 X 108 was the amount of virus in the concentrate. That is an increase of 157%. The results with LASV were similar. Either I didn’t understand what was done or there is something wrong with the math.

We checked and there was indeed a calculation mistake in this paragraph. As requested by reviewer 2, we performed additional experiments to show the viral recovery rate for different amounts of initial virus input and figure 2 has been completely replaced.

Minor:
Line 38: This is not the family name so it shouldn’t be capitalized.
Thank you very much, we corrected the spelling.
Line 39: Please change annually to annual.
Thank you very much, we corrected the spelling.
Line 30 & 40: Be consistent with the shorthand virus names Guanirito virus and Sabiá virus.

Thank you for your comment. Even though it is confusing that in one sentence some viruses are mentioned with their full and shorthand virus name and others only with their spelled-out name, we would prefer to introduce the shorthand virus name only if it is used afterwards.

Line 75: Please indicate how the virus stock was validated, for instance sequencing, and or a reference to how the virus was identified.

The MORV strain used for this study has been isolated and sequenced at BNITM and this is described in the indicated reference (Gunther, S., et al., Mopeia virus-related arenavirus in natal multimammate mice, Morogoro, Tanzania. Emerg Infect Dis, 2009. 15(12): p. 2008-12.). The sequence is also deposited in the Gene database (https://www.ncbi.nlm.nih.gov/gene/?term=Morogoro%203017/2004). The strain has since then been regularly checked by partial Sanger sequencing as part of our routine quality control measures. We changed the relevant sentence in the manuscript to better reflect that the specific strain has been characterized in the indicated publication. The sentence now reads: “MORV strain 3017/2004 had been isolated and sequenced in our laboratory [14] and the used stock was passaged ≤ 3 times.“

Line 134: Please chnnge volumn to volume.

Thank you very much, we corrected the spelling.

Line 225: Please change Formaldehyde to formaldehyde

Thank you very much, we corrected the spelling.

In summary, we have responded to all comments and have modified the manuscript accordingly. We thank the reviewers for the constructive criticism and hope that the revised version is now fully acceptable for publication in Viruses.

Best regards,

Lisa Oestereich

Reviewer 2 Report

The manuscript "Validation of inactivation methods for arenaviruses" by authors Olschewski et al. reports on the evaluation of an ultrafiltration protocol to remove cytotoxic components from common virus inactivation buffers and chemicals prior to using cell culture to validate the inactivation of a proxy arenavirus in culture supernatant or cells. The authors show that all the chemical and physical inactivation methods was able to reduce virus titres to below detection level.

My major concern with the study is the use of the Amicon Ultra-15 100K filter devices to concentrate virus and remove cytotoxic chemicals. The authors did perform a verification of virus recovery, however this was not done appropriately and since this is the foundation the rest of the manuscript is built on, I would urge the authors to do further verification of virus recovery, from lower titre input samples, following ultrafiltration.  

The authors claim that the recovery rate was 85% and that the device is well suited for the purpose of this study. However, the major caveat is that the authors only evaluated the recovery from samples with high viral titres in the input material (between 1.4 x 10^6 to 1.5 x 10^6 FFU/ml in the input material). The authors did not evaluate recovery from low titre input samples. In practice, after inactivation of a sample, on would expect a drop in replicating virus titre in a sample. This could be a complete inactivation, but in some cases it might only result in a drop in titre. From personal experience with a recent experiment, recovery of virus from low titre samples is much poorer with Amicon-15 100K devices than recovery from high titre samples. Therefore, by using these filtration devices on samples that were subjected to inactivation protocols, the authors might have missed some residual amounts of virus left in samples, that were then lost following ultrafiltration.

The authors should repeat and expand the virus recovery experiments, by preparing samples with a decreasing amount of input virus before ultrafiltration, and present those results.

Specific comments:

line 117: the authors describe splitting inoculated cell cultures 1:10 once or twice a week. The authors state that they kept the medium from the parent flask and added it at a 1:2 ratio to fresh medium in new flasks. It is not clear if the authors kept all the cells when splitting. For example when splitting a T25cm flask 1:10, did they end up with 10 new flasks, or did they only take a tenth of the cells and place it in the new flask, and discarded 90% of the cells from the parent flask? If so, what are the chances, with a low level of infection, that the authors might have inadvertently discarded some infected cells?

line 189: the authors refer to a "falcon". Falcon is a brand name of tubes. Did they authors specifically use the Falcon brand, or do they mean they used a 15ml conical centrifuge tube?

Author Response

Please find enclosed our reply to the comment from Reviewer 2. All changes in the manuscript are highlighted in yellow (named “viruses-1187493_changes highlighted”) and one additional Appendix A has been added. To comply with the requests of reviewer 2 we performed additional experiments and figure 2 has consequently been replaced.

The manuscript "Validation of inactivation methods for arenaviruses" by authors Olschewski et al. reports on the evaluation of an ultrafiltration protocol to remove cytotoxic components from common virus inactivation buffers and chemicals prior to using cell culture to validate the inactivation of a proxy arenavirus in culture supernatant or cells. The authors show that all the chemical and physical inactivation methods was able to reduce virus titres to below detection level.

My major concern with the study is the use of the Amicon Ultra-15 100K filter devices to concentrate virus and remove cytotoxic chemicals. The authors did perform a verification of virus recovery, however this was not done appropriately and since this is the foundation the rest of the manuscript is built on, I would urge the authors to do further verification of virus recovery, from lower titre input samples, following ultrafiltration.

The authors claim that the recovery rate was 85% and that the device is well suited for the purpose of this study. However, the major caveat is that the authors only evaluated the recovery from samples with high viral titres in the input material (between 1.4 x 10^6 to 1.5 x 10^6 FFU/ml in the input material). The authors did not evaluate recovery from low titre input samples. In practice, after inactivation of a sample, on would expect a drop in replicating virus titre in a sample. This could be a complete inactivation, but in some cases it might only result in a drop in titre. From personal experience with a recent experiment, recovery of virus from low titre samples is much poorer with Amicon-15 100K devices than recovery from high titre samples. Therefore, by using these filtration devices on samples that were subjected to inactivation protocols, the authors might have missed some residual amounts of virus left in samples, that were then lost following ultrafiltration.

Member of the Leibniz Association
National Reference Centre for Tropical Pathogens
WHO Collaborating Centre for Arbovirus and Haemorrhagic Fever Reference and Research www.bnitm.de

Tel Fax E-Mail

The authors should repeat and expand the virus recovery experiments, by preparing samples with a decreasing amount of input virus before ultrafiltration, and present those results.

Thank you for your comment. We agree that the level of viral titer has an impact on the recovery rate and have performed additional experiments to evaluate the recovery rate in dependency on the input amount. We measured both the recovery of viral RNA and infectious particles to distinguish between the total loss of particles (loss of RNA & infectious virus) and the reduction of infectivity due to the treatment (loss of infectious particles > loss of viral RNA). We tested input amounts from 106 to 100 FFU. The results of these experiments are depicted in the new Figure 2.

We analysed the input, the concentrate, and the flow-through of all samples with IFA and real-time RT-PCR. The inputs and the concentrates were additionally used to infect Vero cells to check for virus replication with indirect immunofluorescence staining of the viral nucleoprotein (not shown in the publication).

Even at the lowest input amount (100 FFU), the three replicates of the ultra-filtrated sample showed a positive result in indirect immunofluorescence and infectious virus could be detected with IFA. Overall, our recovery was above 50 % for all tested dilutions.

We added the information about the real-time RT-PCR measurement of MORV in Appendix A and modified Figure 2. The paragraph 2.2 has been changed to: “The recovery rate of infectious arenaviruses with the Amicon®Ultra-15 100K Centrifugal Filter Device (Millipore, USA) as concentrator was evaluated for MORV. Fifty ml of viral stocks ranging from high to low amounts of infectious particles (106 to 100 FFU) were centrifuged through the device at room temperature (RT) until the volume was reduced to 2 ml in triplicates according to the manufacture’s specifications. In three in-dependent experiments 50 ml virus containing cell culture supernatant with known viral titers were centrifuged at 2500 x g and room temperature (RT) through the device until the volume was reduced to 2 ml. The amount of infectious titer viral particles in the concentrate were analyzed by IFA to determine the recovery rate. Additionally, the amount of viral particles in the input and the flow- through were determined. Viral RNA in input, concentrate, and flow-through was quantified with MORV-specific real-time RT-PCR (see Appendix A).”

The results part 3.1 has been changed to: “To assess the impact of the virus concentration in the starting material on the recovery rate, we performed ultrafiltration with different input amounts of MORV. The numbers of infectious particles ranged from 106 to 100 FFU and the experiments were done in triplicates. The mean recovery rate of infectious MORV was found to be 56 – 106% and the recovery rate of the viral RNA ranged from 49 – 117% (Figure 2). Lower recovery rates were generally observed for the lower input amounts. However, we still achieved more than 50% mean recovery at the lowest tested input (100 FFU). Therefore, ultrafiltration with this device is well suited to concentrate arenaviruses and was used in this study to remove cytotoxic substances after inactivation.”

Specific comments:

line 117: the authors describe splitting inoculated cell cultures 1:10 once or twice a week. The authors state that they kept the medium from the parent flask and added it at a 1:2 ratio to fresh medium in new flasks. It is not clear if the authors kept all the cells when splitting. For example when splitting a T25cm flask 1:10, did they end up with 10 new flasks, or did they only take a tenth of the cells and place it in the new flask, and discarded 90% of the cells from the parent flask? If so, what are the

chances, with a low level of infection, that the authors might have inadvertently discarded some infected cells?

We decided to split the cells and only keep 1:10 of the cells and 1:2 of the medium. The first cell splitting in this set-up was done at least 3 days post initial inoculation with the concentrates. Based on our experience with MORV replication in Vero cells (virus rescue, virus isolations from rodent samples, growth kinetics with different MOIs), keeping only 1:2 of the medium and 1:10 of the cells is sufficient to prevent accidental loss of all infectious particles.

In our additional we also inoculated Vero cells with the concentrates (not shown in the publication) and we could show that virus-positive cells (immunofluorescence staining of infected cells) were detectable already 3 days post-inoculation and the rate of infected cells was approximately 50 – 80 % for the lowest tested input. We think that the chances are very low, that the dilutions used for the assay would lead to false-negative results in all three replicates.

line 189: the authors refer to a "falcon". Falcon is a brand name of tubes. Did they authors specifically use the Falcon brand, or do they mean they used a 15ml conical centrifuge tube?

Thank you for bringing this to our attention, we have changed it to conical centrifuge tube.

In summary, we have responded to all comments and have modified the manuscript accordingly. We thank the reviewers for the constructive criticism and hope that the revised version is now fully acceptable for publication in Viruses.

Best regards,

Lisa Oestereich

Round 2

Reviewer 1 Report

Please check the references.  Reference [20] was inserted in the text but not included in the reference list.  Check all the references after [20] to be sure that they match with the citations in the text after the inclusion of [20] in the reference list.  Other than that a very pleasurable paper to read.  Thank you for clarifying the limitations of the study.

Author Response

Please find enclosed our reply to the comment from Reviewer 1. All changes in the manuscript are highlighted in track changes mode (named “viruses-1187493-1-revised”).

Please check the references.  Reference [20] was inserted in the text but not included in the reference list.  Check all the references after [20] to be sure that they match with the citations in the text after the inclusion of [20] in the reference list.  Other than that a very pleasurable paper to read.  Thank you for clarifying the limitations of the study.

We checked all references and added the missing paper.

We thank the reviewers for the constructive criticism and hope that the revised version is now fully acceptable for publication in Viruses.

Best regards,

Lisa Oestereich

Reviewer 2 Report

I would like to thank the authors for performing the additional laboratory work to qualm concerns about virus recovery from low titre samples. 

Please check the formatting of the new Figure 2. It appears (in the PDF version of the manuscript I was able download for review), that there is a problem with the text characters or symbols in the table below the figure. Some text are displayed as squares. The formatting of the graph x-axis titles is also not correct.

Author Response

Please find enclosed our reply to the comment from Reviewer 2. All changes in the manuscript are highlighted in track changes mode (named “viruses-1187493-1-revised”).

I would like to thank the authors for performing the additional laboratory work to qualm concerns about virus recovery from low titre samples. 

Please check the formatting of the new Figure 2. It appears (in the PDF version of the manuscript I was able download for review), that there is a problem with the text characters or symbols in the table below the figure. Some text are displayed as squares. The formatting of the graph x-axis titles is also not correct.

We carefully checked the manuscript and the figures during the reformation and final spell correcting step and the formation in the Word document appears to be ok.  

We thank the reviewers for the constructive criticism and hope that the revised version is now fully acceptable for publication in Viruses.

Best regards,

Lisa Oestereich
